# Autumn – winter minimum temperature changes in the southern Sikhote-Alin mountain range of northeast Asia since 1529 AD

Olga N. Ukhvatkina, Alexander M. Omelko, Alexander A. Zhmerenetsky, Tatyana Y. Petrenko.

Federal Scientific center of the East Asia terrestrial biodiversity Far Eastern Branch of Russian Academy of Sciences, Vladivostok    690022    RUSSIA

*Correspondence to:* Olga Ukhvatkina (ukhvatkina@gmail.com)

**Abstract.** The aim of our research was to reconstruct climatic parameters (for the first time for the Sikhote-Alin mountain range) and to compare them with global climate fluctuations. As a result, we have found that one of the most important limiting factors for the study area is the minimum temperatures of the previous autumn-winter season (August-December), and this finding perfectly conforms to that in other territories. We reconstructed the previous August-December minimum temperature for 485 years, from 1529 to 2014. We found twelve cold periods (1535-1540, 1550-1555, 1643-1649, 1659-1667, 1675-1689, 1722-1735, 1791-1803, 1807-1818, 1822-1827, 1836-1852, 1868-1887, 1911-1925) and seven warm periods (1560-1585, 1600-1610, 1614-1618, 1738-1743, 1756-1759, 1776-1781, 1944-2014). These periods correlate well with reconstructed data for the Northern Hemisphere and the neighboring territories of China and Japan. Our reconstruction has 3, 9, 20 and 200-year periods, which are may be in line with high-frequency fluctuations in ENSO, the short-term solar cycle, PDO fluctuations and the 200-year solar activity cycle, respectively. We suppose that the temperature of North Pacific, expressed by Pacific Decadal Oscillation may make a major contribution to regional climate variations. We also assume that the regional climatic response to solar activity becomes apparent in the temperature changes in the northern part of Pacific Ocean and corresponds to cold periods during the solar minimum. These comparisons show that our climatic reconstruction based on tree-ring chronology for this area may potentially provide a proxy record for long-term, large-scale past temperature patterns for northeast Asia. The reconstruction reflects the global traits and local variations in the climatic processes of the southern territory of the Russian Far East for more than the past 450 years.

## 1 Introduction

Global climate change is the main challenge for human life and natural systems, which is why we should clearly understand climatic changes and their mechanisms. A retrospective review of climatic events is necessary for understanding the climatic conditions from a long-term perspective. At the same time, instrumental climate observations rarely cover more than a 100-year period and are often restricted to 50-70 years. This restriction forces the researchers to continuously find new ways and methods to reconstruct climatic fluctuations. Dendrochronology has been widely applied in climatic reconstruction for local territories and at the global scale for both climatic reconstructions of the past few centuries and paleoclimatic reconstructions because it is rather precise, extensively used and a replicable instrument (Corona et al.; Popa and Bouriaud, 2014; Kress et al., 2014; Lyu et al., 2016).

A great number of studies have focused on climatic change reconstruction for the northeastern parts of China based on *P. koraeinsis* radial growth studies (e.g., Zhu et al., 2009; Wang et al., 2013; Wang et al., 2016; Zhu et al., 2015; Lyu et al. 2016). Climatic parameters were reconstructed for the whole Northern Hemisphere (Wilson et al., 2016), China (Ge et al., 2016), and temperature characteristics were reconstructed for northeastern Asia (Ohyama et al., 2013). Despite this, there are very few studies of Russian Far East climate (e.g., Willes et al., 2014; Jacoby et al.,

2004; Shan et al., 2015); moreover, there is an absence of dendrochronological studies for the continental part of Russian Far East. Meanwhile, most of species present in northeastern China, the Korean peninsula and Japan grow in this region. In addition, the distribution areas of these trees often end in the south of the Russian Far East, which increases the climatic sensitivity of plants. Additionally, some parts of the forests in the Russian Far Eastern have not been subjected to human activity for the last 2000-4000 years. This makes it possible to forests extend the studied timespan. In addition, the southern territory of the Russian Far East is sensitive to global climatic changes as it is under the influence of cold air flow from northeastern Asia during the winter and summer monsoons. All of the factors listed above create favorable conditions for dendroclimatic studies.

It is well-known that cold and warm periods of the climate is correlated with intensive solar activity (e.g., the Medieval Warm Period), while decreases in temperature occurs during periods of low solar activity (e.g., the Little Ice Age; Lean and Rind, 1999; Bond et al., 2001). According to findings from an area of China neighboring the territory studied here, the registered warming has been significantly affected by global warming since the 20th century (Ding and Dai, 1994; Wang et al., 2004; Zhao et al., 2009), which is often indicated by a faster rise in night or minimum temperatures (Karl et al., 1993; Ren and Zhai, 1998; Tang et al., 2005). To better understand and evaluate future temperature change trends, we should study the long-term history of climatic changes.

However, using tree-ring series for northeastern Asia (particularly temperature) is rather complicated due to the unique hydrothermal conditions of the region. Most reconstructions cover periods of less than 250 years (e.g., Shao and Wu, 1997; Zhu et al., 2009; Wang et al., 2012; Li and Wang, 2013; Yin et al., 2009; Zhu et al., 2015), except for a few with periods up to 400 years (Lyu et al., 2016; Wiles et al., 2014). The short period of reconstructions is the reason why such reconstructions cannot capture low-frequency climate variations.

The warming of the climate (particularly minimum temperature increase) is registered across the whole territory of northeastern Asia (Lyu et al., 2016). In the Russian Far-East, such warming has been recorded for more than 40 past years (Kozhevnikova, 2009). However, the lack of detailed climatic reconstructions for the last few centuries makes it difficult to capture long-period climatic events for this territory and interpret the temperature conditions for the last 500-1000 years.

Therefore, the main objectives of this study were (1) to develop the first three-ring-width chronology for the southern part of the Russian Far East; (2) to analyze the regime of temperature variation over the past centuries in the southern part of the Russian Far East; (3) to identify the recent warming amplitude in context of long-term changes and to analyze the periodicity of climatic events and their driving forces. Our new minimum temperature record supplements the existing data for northeast Asia and provides new evidence of past climate variability. There is the potential to better understand future climatic trajectories from these data in northeast Asia.

## 2 Materials and methods

### 2.1 Study area

We studied the western macroslope of the southern part of the Sikhote-Alin mountain range (Southeastern Russia) at the Verkhneussuriysky Research Station of the Federal Scientific Center of the East Asia terrestrial biodiversity Far East Branch of the Russian Academy of Sciences (4400 ha; N 44°01'35.3'', E 134°12'59.8'', Fig. 1).

The territory is characterized by a monsoon climate with relatively long, cold winters and warm, rainy summers. The average annual air temperature is 0.9 °C; January is the coldest month (–32 °C average temperature), and July is the warmest month (27 °C average temperature). The average annual precipitation is 832 mm (Kozhevnikova, 2009). Southerly and southeasterly winds predominate during the spring and summer, while northerly and northwesterly

winds predominate in autumn and winter. The terrain includes mountain slopes with an average angle of $\sim 20°$, and
the study area is characterized by brown mountain forest soils (Ivanov, 1964) (Fig. 2).
Mixed forests with Korean pine (*Pinus koraensis* Siebold et Zucc.) are the main vegetation type in the study area,
and they form an altitudinal belt up to 800 m above sea level. These trees are gradually replaced by coniferous fir-
spruce forests at high altitudes (Kolesnikov, 1956). Korean pine-broadleaved forests are formed by up to 30 tree
species, with *Abies nephrolepis* (Trautv.) Maxim, *Betula costata* (Trautv.) Regel*., Picea jezoensis* (Siebold et Zucc.)
Carr., *P. koraeinsis* and *Tilia amurensis* Rupr. being dominant.
Korean pine-broadleaved forests are the main forest vegetation type in the Sikhote-Alin mountain range in the
southern part of the Russian Far East. This area is the northeastern limit of the range of Korean pine-broadleaved
forests, which are also found in northeastern China (the central part of the range), on the Korean peninsula, and in
Japan. The Sikhote-Alin mountain range is one of the few places where significant areas of old-growth Korean pine-
broadleaved forest remain. In the absence of volcanic activity, which is a source of strong natural disturbances in the
central part of the range (Liu, 1997; Ishikava, 1999; Dai et al., 2011), wind is the primary disturbance factor on this
territory. Wind causes a wide range of disturbance events, from individual treefalls to large blowdowns (Dai et al.,

97    2011).

Approximately 60% of the Research Station area had been subjected to selective clear-cutting before the station was
established in 1972. The remaining 40% of its area has never been clear-cut and is covered by unique old-growth
forest.

### 2.2 Tree-ring chronology development

Our study is based on data collected in a 10.5-ha permanent plot (Omelko and Ukhvatkina, 2012; Omelko et al., 2016),
which was located in the middle portion of a west-facing slope with an angle of $22°$ at a gradient altitude 750-950 m
above sea level. The forest in the plot was a late-successional stand belonging to the middle type of Korean pine-
broadleaved forests at the upper bound of the distribution of Korean pine, where it forms mixed stands of Korean
pine-spruce and spruce-broadleaved forests (Kolesnikov, 1956).
One core per undamaged old-growth mature tree (25 cores from 25 trees) and one sample from dead trees (20 samples)
were extracted from *P. koraiensis* trees in the sample plots from the trunks at breast height. In the laboratory, all tree-
ring samples were mounted, dried and progressively sanded to a fine polish until individual tracheids within annual
rings were visible under an anatomical microscope according to standard dendrochronological procedures (Fritts,
1976; Cook and Kairiukstis, 1990). Preliminary calendar years were assigned to each growth ring, and possible errors
in measurement due to false or locally absent rings were identified using the Skeleton-plot cross-dating method
(Stokes and Smiley, 1968). The cores were measured using the semi-automatic Velmex measuring system (Velmex,
Inc., Bloomfield, NY, USA) with a precision of 0.01 mm. Then, the COFECHA program was used to check the
accuracy of the cross-dated measurements (Holmes, 1983). To mitigate the potential trend distortion problem in
traditionally detrended chronology (Melvin and Briffa, 2008; Anchukaitis et al., 2013), we used a signal-free method
(Melvin and Briffa, 2008) to detrend the tree-ring series using the RCSigFree program (http://www.ldeo.
columbia.edu/tree-ring-laboratory/resources/software).
Age-related trends were removed from the raw tree-ring series using an age-dependent spline smoothing method. The
ratio method was used to calculate tree-ring indices, and the age-dependent spline was selected to stabilize the variance
caused by core numbers. Finally, the stabilized signal-free chronology was used for the subsequent analysis (Fig. 3).
The mean correlations between trees (*R*bt), mean sensitivity (MS) and expressed population signal (EPS) were
calculated to evaluate the quality of the chronology (Fritts, 1976). *R*bt reflects the high-frequency variance, and MS
describes the mean percentage change from each measured annual ring value to the next (Fritts, 1976; Cook and
Kairiukstis, 1990). EPS indicates the extent to which the sample size is representative of a theoretical population with
an infinite number of individuals. A level of 0.85 in the EPS is considered to indicate a chronology of satisfactory
quality (Wigley *et al.*, 1984). The statistical characteristics of the chronology are listed in Table 1.
The full length of the chronology spans (VUS chronology) from 1451 to 2015. A generally acceptable threshold of
the EPS was consistently greater than 0.85 from AD 1602 to 2015 (9 trees; Fig. 3b), which affirmed that this is a
reliable period. However, although the EPS value from AD 1529 to 1602 was less than 0.85, it matches a minimum
sample depth of 4 trees in this segment (EPS>0.75). Although the record from AD 1529 to 1602 is thus less certain,
we here report it as it is very important to extend the tree-ring chronology as much as possible because there are only
a few long climate reconstructions in this area. Therefore, we retained the part from 1529 to 1602 in the reconstruction.
**2.3 Climate data and statistical methods**
Monthly precipitation, monthly mean and minimum temperature data were obtained from the Chuguevka
meteorological station (44.151462 N, 133.869530 E, about 30 km from Verkhneussuriisky research station) and the
meteorological post at the Verkhneussuriisky research station of the Federal Scientific Center of the East Asia
terrestrial biodiversity FEB RAS (Meteostation 7 – MP7) as well. The periods of monthly data available from the
Chuguevka and Verkhneussuriisky stations are 1936-2004 and 1969-2004, respectively (1971-2003 for minimum
temperature data from the Chuguevka).
The data of large-scale climate conditions, such as the Northern Hemisphere temperature (NH), North Atlantic
Oscillation (AMO), Pacific Decadal Oscillation (PDO) and Nino3 reconstruction (Mann et al., 2009), and also
indicators of solar activity, such as reconstructed solar constant (TSI, Lean, 2000) and sun spot number (SSN) were
downloaded and analyzed in Royal Netherlands Meteorological Institute climate explorer (http://climexp.knmi.nl).
To demonstrate that our reconstruction representative and reflect temperature variations, we conduced spatial
correlation between our temperature reconstruction and gridded temperature dataset of the Climate Research Unit
(CRU TS4.00) for the period 1960-2003, by using the Royal Netherlands Meteorological Institute climate explorer
(http://climexp.knmi.nl).
**2.4 Statistical analyses**
A correlation analysis was used to evaluate the relationships between the ring-width index and observed monthly
climate records from the previous June to the current September. To identify the climate-growth relationships of
Korean pine in the southern Sikhote-Alin mountain range, a Pearson's correlation was performed between climate
variables and tree-width index. We used a traditional split-period calibration/verification method to explore the
temporal stability and reliability of the reconstruction model (Fritts, 1976; Cook and Kairiukstis, 1990). The Pearson's
correlation coefficient ($r$), R-squared ($R^2$), the redaction of the error (RE) the coefficient of efficiency (CE), and the
product means test (PMT) were used to verify the results. Analyses were carried out in R using the treeclim package
(Zang and Biondi, 2015) and STATISTICA software (StatSoft®). Analyses of reconstruction included multi-taper
method (MTM) (Mann & Lees, 1996) and Monte Carlo Singular Spectrum Analysis (SSA; Allen and Smith, 1996).
Analysis was carried out in SSA-MTM Toolkit for Spectral Analysis software (Ghil et al., 2001; Dettinger et al.,
160 1995).

**3 Results**

**3.1 Climate-radial growth relationship**

Relationships between the VUS chronology and monthly climate data are shown in Fig 4. To reveal the correlation
between climatic parameters and radial growth change of *P. koraiensis,* we had three data sets: the first-time series
had a length of 68 years (1936-2004, Chuguevka), the second had a length of 34 years (1966-2000, MP7), and the
third had a length of 33 years (1971-2003, Chuguevka, minimum temperature). To select the appropriate parameters,
we analyzed all datasets. As a result, we revealed a reliable but slight positive correlation between *P. koraiensis* growth
and precipitation in May and June of the current year and September of the previous year in the territory of Chuguevka
village (Fig. 4a). There is also a slight positive correlation with precipitation in September of the previous year and
May of the current year at Metheostation 7 (MP7) (Fig. 4b). In addition, we revealed a slight negative correlation with
precipitation in February-March of the current year.
As for the correlation between temperature and *P. koraiensis* growth, the analysis reveals a weak positive correlation
with the average monthly temperature in June of the previous year and in February-April of the current year in the
Chuguevka settlement and a slight negative correlation with the average monthly temperature in June-July as well
(Fig. 4c). The analysis of the correlation with the average monthly temperature at Metheostation 7 (MP7) shows us a
weak positive correlation with temperature in August and December of the preceding year and a negative correlation
with temperature in July of the current year (Fig. 4d). In addition, we analyzed the correlation with minimum average
monthly temperatures at MP7 and Chuguevka. The revealed correlation with minimum temperature is reliable but
weak (Fig. 4e,f).
Moreover, based on the weak interaction that was revealed, we analyzed the correlation with climatic parameters for
selected ranges of months (Fig. 4h,g). The highest significant correlation appears between growth and the minimum
monthly temperature of August-December of the previous year at Chuguevka (Fig. 4h), on which we base our
subsequent reconstructions.

**3.2 Minimum temperature reconstruction**

Basing on analysis of the correlation between climatic parameters and Korean pine growth, we constructed a linear
regression equation to reconstruct the minimum monthly temperature of August-December of the previous year
(VUSr). The transfer function was as follows:
$VUSr = 7.189 X_t - 15.161$
($N=32$, $R=0.620$, $R^2=0.385$, $R^2_{adj}=0.364$, $F=18.76$, $p < 0.001$)
where *VUSr* is the August-December minimum temperature at Chuguevka and *X* is the tree-ring index of the Korean
pine RSC chronology in year *t*. The comparison between the reconstructed and observed mean growing season
temperatures during the calibration period is shown in Fig. 5(a). The cross-validation test for the calibration period
(1971-1997, R=0.624) yielded a positive RE of 0.334, a CE of 0.284, and the cross-validation test for calibration
period 1977-2003 (R=0.542) a positive RE of 0.654, a CE 0f 0.644, confirming the predictive ability of the model.
Although during the study period, the model shows the observed values very well, the short observation period (1971-
2003) does not allow using split-sampling calibration and verification methods in full for evaluating quality and model
stability. This limitation is why we used a bootstrapping resampling approach (Efron, 1979; Young, 1994) for stability
evaluation and transfer function precision. The idea that this method is based on indicates that the available data
already include all the necessary information for describing the empirical probability for all statistics of interest.
Bootstrapping can provide the standard errors of statistical estimators even when no theory exists (Lui et al., 2009).
The calibration and verification statistics are shown in Table 2. The statistical parameters used in bootstrapping are
very similar to those from the original regression model, and this proves that the model is quite stable and reliable and
that it can be used for temperature reconstruction.

**3.3 Temperature variations from AD 1529 to 2014 and temperature periodicity**
Variations in the reconstructed average minimum temperature of the previous August-December (VUSr) since AD
1529 and its 21-year moving average are shown in Fig. 5b. The 21-year moving average of the reconstructed series
was used to obtain low-frequency information and analyze temperature variability in this region. The mean value of
the 486-year reconstructed temperature was -7.93° C with a standard deviation of ±1.40° C. We defined warm and
cold periods as when temperature deviated from the mean value plus or minus 0.5 times the standard deviation,
respectively (Fig. 5b). If the reconstructed minimum temperatures were above or below the average value by >0.5 SD
for three or more years, then we considered this deviation as warm or cold period, respectively. Also, if two warm (or
cold) periods were separated by one year, when the temperature sharply decreased (or increased), then such periods
merged into one.
Hence, warm periods occurred in 1560-1585, 1600-1610, 1614-1618, 1738-1743, 1756-1759, 1776-1781, 1944-2014,
and cold periods appeared in 1535-1540, 1550-1555, 1643-1649, 1659-1667, 1675-1689, 1722-1735, 1791-1803,
1807-1818, 1822-1827, 1836-1852, 1868-1887, 1911-1925. Among them, the four warmest years were in 1574 (-
4.35° C), 1606 (-5.35° C), 1615 (-5.71° C), 1741 (-5.36° C), 1757 (-6.16° C), 1779 (-5.21° C), 2008 (-2.72° C), while
the three coldest year were in 1543 (-9.84° C), 1551 (-9.88° C), 1647 (-10.77° C), 1662 (-11.10° C), 1685 (-9.45° C),
1728 (-10.08° C), 1799 (-10.70° C), 1815 (-10.13° C), 1825 (-9.87° C), 1843 (-10.55° C), 1883 (-10.73° C), 1913 (-
10.29° C). The longest cold period extended from 1868 to 1887, and the longest warm period extended from 1944 to
present day. The coldest year is 1662 (-11.10° C) and the warmest year is 2008 (-2.72° C).
The MTM spectral analysis over the full length of our reconstruction revealed significant ($p < 0.05$) cycle peaks at
2.3-year (95%), 2.5-year (99%), 2.9-year (99%), 3.0-year (99%), 3.3-year (95%), 3.7-year (95%), 8.9-year (99%)
short periods and 20.4-year (95%), 47.6-year (95%), 188.7-year (99%) long periods (Fig. 6). Singular spectrum
analysis (SSA) reveals 8 leading temporal modes that significant at the 95% confidence level (Allen & Smith, 1996).
Of these, SSA analysis reveals a single significant low order mode variability near 200 years, but there is little evidence
in the reconstruction variability at the 40-50 years. Also 3 significant power periods were reveals: 20.4-year, 9-year
and near 3-year periods. Comparison of the reconstruction and global temperature for oceans of Northern Hemisphere
(NH), North Atlantic Oscillation (AMO), Pacific Decadal Oscillation (PDO) and Nino3 reconstruction (Mann et al.,
2009) show significant correlation between reconstruction and NH ($r$=0.67, $p$<0.0001), AMO ($r$=0.49, $p$<0.001), and
PDO ($r$=0.68, $p$<0.0001), and non-significant correlation between reconstruction and Nino3 reconstruction ($r$=0.27,
$p$=0.08). Comparison of the reconstruction and indicators of solar activity shows significant correlation of the
minimum temperature with the TSI ($r$=0.52, $p$<0.0001) and non-significant correlation with SSN ($r$=0.26, $p$<0.1).
Comparison of the instrumental climate data and instrumental indicators of solar activity shows significant correlation
of the minimum temperature with the TSI ($r$=0.52, $p$<0.0001) and non-significant correlation with SSN ($r$=0.26,
$p$<0.1).
Spatial correlations between our reconstruction and the CRU TS4.00 temperature dataset reveal our record's
geographical representation (Fig. 7). The results show that the reconstruction of mean minimum temperature of
previous August – December is significantly positively correlated with the CRU TS4.00 (r=0.568, p<0.0001).

**4 Discussion**

**4.1 Climate-growth relationships**

The results of our analysis suggest that the radial growth of Korean pine in the southern part of the Sikhote-Alin mountain range is mainly limited by the pre-growth autumn-winter season temperatures, in particular the minimum temperatures of August-December (Fig. 4). It is widely known that tree-ring growth in cold and wet ecotopes, situated on sufficiently high elevation in the Northern Hemisphere, strongly correlate with temperature variability in large areas of Asia, Eurasia, North America (Zhu et al., 2009; Anchukaitis et al., 2013; Thapa et al., 2015; Wiles et al., 2014). The limiting influence of temperature on *P. koraiensis* growth has been mentioned in many studies (Wang et al., 2016; Yin et al., 2009; Wang et al., 2013; Zhu et al., 2009). However, the temperature has various limiting effects in different conditions, and these limiting effects manifest in different ways (Wang et al., 2016). For example, Zhu et al., 2016 indicates that in more northern and arid conditions of the Zhangguangcai Mountains, while precipitation is not the main limiting factor, precipitation is considerably below evaporation during the growth season. This finding is why a stable correlation between *P. koraiensis* growth and the growth season temperature is revealed. This finding is also why moisture availability in soil might be the main limiting factor for Korean pine growth (Zhu et al., 2016), but the emergence of this circumstance can be different in different conditions.

The correlation between growth and minimum temperatures in August-December of the previous year, as revealed in our research, was also mentioned for Korean pine in other works (Wang et al., 2016; Zhang et al., 2015). This finding may be explained by the following circumstances. Extreme temperatures limit the growth of trees at the tree line or in high-latitude forests (Wilson and Luckman, 2002; Körner and Paulsen, 2004; Porter et al., 2013; Yin et al., 2015). Taking into consideration the fact that the study area is situated at the altitudinal limit of Korean pine forest distributions, in particular the Korean pine (Kolesnikov, 1956), these findings seem to be reliable.

In addition, in the conditions close to extreme for this species, low temperatures in autumn-winter may lead to thicker snow cover, which melts far more slowly in spring (Zhang et al., 2015). The study area is notable for its dry spring, and the amount of precipitation is minimal during the most important period of tree-growth in April-May (Kozhevnikova, 2009). If the vegetation period of the plant cannot begin at the end of March and packed snow cover melting is impeded up until the beginning of May, plant growth may be reduced. Moreover, although cambial activity stops in the winter, organic components are still synthesized by photosynthesis. Low temperatures (in the territory of the VUSr it can reach -48°C in certain years) may induce to loss of accumulated materials, which adversely affects growth (Zhang et al., 2015). The study area is in the center of the vegetated area, where the conditions for Korean pine growth are optimal during the growing season, and only minimum temperature is regarded as an extreme factor.

**4.2 Comparison with other tree-ring-based temperature reconstructions**

At present, temperature reconstructions are uncommon for the Russian Far East, and research sites are located for thousands of kilometers away from one another. For example, Wiles et al. undertook a study of summer temperatures on Sakhalin Island (Wiles et al., 2014). Unfortunately, it is impossible to compare our findings with theirs because Sakhalin Island is climatically far more similar to Japanese islands than to the Sikhote-Alin mountains, and temperature variations in their study area are mainly caused by oceanic currents.

In addition, instrumental observations from the study area rarely encompass a period longer than 50 years (and studies have only been conducted for large settlements). Consequently, the tree-ring record serves as a good indicator of the past cold-warm fluctuations in the Russian Far East. The analysis of spatial correlations between our reconstruction

and the CRU TS4.00 temperature dataset reveal spatial correlations between the observed and reconstruction
minimum temperatures from the CRU TS4.00 gridded $T_{min}$ dataset during the baseline period of 1960-2003 (Fig. 7).
It's indicating that our temperature reconstruction is representative of large-scale regional temperature variations and
can be taken as representative of southeastern of the Russian Far East and northeastern of the China.
To identify the regional representativeness of our reconstruction, we compared it with two temperature reconstructions
for surroundings areas (Fig. 1) and a reconstruction for the Northern Hemisphere (Fig. 8). The first reconstruction was
for summer temperatures in the Northern Hemisphere (Wilson et al., 2016; Fig. 1). The second reconstruction was an
April-July tree-ring-based minimum temperature reconstruction for Laobai Mountain (northeast China), which is
approximately 500 km northwest of our site. The third was a February-April temperature reconstruction for the
Changbai Mountain (Zhu et al., 2009; Fig. 1), which are approximately 430 km southwest of our site. Although the
spring and summer temperatures have been reconstructed in the last two cases, we use these reconstructions for
comparison, because, firstly, there are no other reconstructions for this region, and secondly, despite the possible
seasonal shifts, long cold and warm periods should be identified in all seasons.
Cold and warm periods are shown in table 3 (the duration is given by the authors of the article). The reconstructions
show that practically all cold and warm periods coincide but have different durations and intensities. The data on
Northern Hemisphere show considerable overlaps of cold and warm periods, and the correlation between
reconstructions is 0.45 ($p > 0.001$). At the same time, we found the warm period 1560-1585, which is not clearly
shown in reconstruction for the Northern Hemisphere, though the general trend of temperature change is maintained
during this period (Fig. 8). Long cold periods from 1643 to 1667 and 1675-1690 that were revealed for another territory
(Lyu et al., 2016; Wilson et al., 2016) coincided with the Maunder Minimum (1645–1715), an interval of decreased
solar irradiance (Bard et al., 2000). The coldest year in this study (1662) revealed in this period too. The Dalton
minimum period centered in 1810 is also notable. Interestingly that cold periods of 1807-1818, 1822-1827, 1836-1852
and 1868-1887 is also registered in reconstructions for Asia (Ohayama et al., 2013) and by Japanese researchers
(Fukaishi & Tagami, 1992; Hirano & Mikami, 2007). Moreover, instrumental observations reconstructed for western
Japanese territories (the nearest to the study area) provide evidence of a cold period in the 1830s-1880s with a short
warm spell in the 1850s (Zaiki et al., 2006), which is in agreement with our data (not reliably period 1855-1865, Tabl.
3). For this period, there are contemporaneous records of severe hunger in Japan in 1832 and 1839, which was the
result of a summer temperature decrease and rice crop failure (Nishimura & Yoshikawa, 1936).
In this case, the longer cold period for the study area can be explained by the relatively lower influence of the warm
current and monsoon and generally colder climate in the south of the Russian Far East compared with Japanese islands.
The differing opinion about the three cold periods in China in the 17[th], 18[th] and 19[th] centuries (Wang et al., 2003) is
also corroborated by our reconstruction. The cold period in the 19[th] century is even more pronounced than that reported
by Lyu et al., 2016. Moreover, Lyu et al., 2016 corroborate that the ascertained cold period in 19[th] century is more
evident in South China, but it is less clear in the northern territories or has inverse trend. Although the Russian Far
East is further north than the southern Chinese provinces and is closer to the northern part of the country, the marked
monsoon climate likely made it possible to reflect the general cold trend in 19[th] century, which was typical both for
China and the entire Northern Hemisphere. Because of this possible explanation, the cold period in the 19[th] century
for the Changbai Mountains shows up more distinctly than for the northern and western territory of Laobai Mountain
(Fig. 8).
Apparently, this discrepancy in regional climate flow is the reason that our reconstruction agrees well with the general
reconstruction for the whole hemisphere ($r = 0.45$, $p < 0.001$) and to a lesser extent agrees with the regional curves for
Laobai Mountain ($r = 0.23$, $p < 0.001$) and Changbai Mountain ($r = 0.32$, $p < 0.001$).
The changing dynamics of the 20[th] century temperature is also interesting to watch. The comparison of the minimum
annual temperatures for the territory and the reconstructed data for the period of 1960-2003 shows significant data
correlation (Fig. 7), including the northeast part of China. At the same time, for Chinese territory (both for southwest
regions and for more northwestern regions), the warming is apparent only in the last quarter century (Zhu et al., 2009)
or at the end of the 20[th] century (Lyu et al., 2016) (Fig. 8 c,d). This trend, revealed for the southern Sikhote-Alin
mountains (a warm spell since 1944), is corroborated for the whole Northern Hemisphere (Wison et al., 2016) (Fig.
8a,b). The maximum cold period is also corroborated, which we note for the 19[th] century (Fig. 8a,b).
The probable explanation is in the regional climate flow differences in the compared data. The territory of northeastern
China is more continental, though the influence of the Pacific Ocean is also notable. At the same time, the southern
part of the Sikhote-Alin mountains is more prone to the influence of monsoons, as are the Japanese islands. According
to paleoreconstructions, the Little Ice Age occurred in the Northern Hemisphere 600-150 years ago (Borisova, 2014).
The period of landscape formation (vegetation types and altitudinal zonation) for the Sikhote-Alin range during the
transition from the Little Ice Age to contemporary conditions occurred within the last 230 years (Razzhigaeva et al.,
2016). The timeframe of the Little Ice Age is generally recognized as varying considerably depending on the region
(Bazarova et al., 2014). However, it is certain that the Little Ice Age is accompanied by an increase in humidity in
coastal areas of northeast Asia (Bazarova et al., 2014). Thus, in similar conditions on the Japanese islands, the Little
Ice Age was accompanied by lingering and intensive rains (Sakaguchi, 1983), and the last typhoon activity was
registered for the Japanese islands from the middle of the 17[th] century to the end of the 19[th] century (Woodruff et al.,
2009). At the same time, the reconstruction of climatic changes for the whole territory of China for the last 2000 years
(Ge et al., 2016) shows that the cold period lasted until 1920, which correlates with the data we obtained. This timespan
wholly coincides with our data, and we can draw the conclusion that in the southern region of the Sikhote-Alin
mountains, the Little Ice Age ended at the turn of the 19[th] century.
Unfortunately, when comparing temperature, different changes were also observed for some cold and warm years
(Fig. 8). This finding may be attributed to differences in the reconstructed temperature parameters (such as average
value, minimum temperature and maximum temperature) and environmental conditions in different sampling regions.
Recent studies show that the oscillations in the medium, minimum and maximum temperature are often asymmetrical
(Karl et al., 1993; Xie and Cao, 1996; Wilson and Luckman, 2002, 2003; Gou et al., 2008). The global warming over
the past few decades has been mainly caused by the rapid growth of night or minimum temperatures but not maximum
temperatures. Meanwhile, some differences between the reconstructed temperature values were well explained by a
comparison with similar areas.
We can conclude that the analysis shows that the reconstructed data is representative for large-scale regional
temperature variations (Fig. 7). At the same time, some cold and warm periods in our reconstruction and other
neighbored studies do not coincide (Fig. 8), which can be due to the reconstruction of other climatic parameters and
differing environmental conditions. So, we believe that these results can characterize regional climate variations and
provide reliable data for large-scale reconstructions for the northeastern portion of Eurasia, but their use for large-
scale regional reconstructions requires further research.
**4.3 Periodicity of climatic changes and their links to global climate processes**
Among the significant periodicities in the reconstructed temperature detected by the MTM analysis (Fig. 7), some
peaks were singled out: 2.3-year (95%), 2.5-year (99%), 2.9-year (99%), 3.0-year (99%), 3.3-year (95%), 3.7-year
(95%), 8.9-year (99%) short periods and 20.4-year (95%), 47.6-year (95%), and 188.7-year (99%) long periods. SSA
analysis shows significant near 3-year, 9-year, 20.4-year and 200-year periods.
The 3-year cycle may be linked with the El Niño-Southern Oscillation (ENSO). These high-frequency (2-7-year)
cycles (Bradley *et al.*, 1987) have also been found in other tree-ring-based temperature reconstructions in northeast
Asia (Zhu *et al.*, 2009; Li and Wang, 2013; Zhu et al., 2016; Gao et al., 2015). The 2–3-year quasi-cycles may also
correspond to the quasi-biennial oscillation (Labitzke and van Loon, 1999) and the tropospheric biennial oscillation
(Meehl, 1987). Despite the fact that many authors establish linkage between 2-7-year cycles and El Niño-Southern
Oscillation (ENSO) or quasi-biennial oscillation in northeastern Asia, we couldn't find significant correlation between
the August-December minimum temperature reconstruction and Nino3, but the analysis showed significant correlation
between the reconstruction and the temperature of Northern Hemisphere oceans. It probably mean that the temperature
variations are more associated with the influence of PDO than ENSO.
On the decadal timescale analysis showed 20-year cycles which may reflect processes influenced by Pacific Decadal
Oscillation (PDO, Mantua and Hare 2002) variability, which has been found at 15-25-yr and 50-70-yr cycles (Ma,
2007). Our analysis shows a significant correlation ($r$=0.68, $p$<0.0001) between reconstruction and the mean annual
PDO index of Mann et al. (2009) from 1900-2000. Taking into account that many researchers, who studied on the
territory of northeast Asia have also revealed these cycles in relation to the Korean pine, we hypothesize that the
Korean pine tree-ring series support the concept of long-term, multidecadal variations in the Pacific (e.g., D'Arrigo
et al., 2001; Cook, 2002; Jacoby et al., 2004; Liu et al., 2009; Li, Wang, 2013; Willes et al., 2014; Lu, 2016) and that
such variation or shifts have been present in the Pacific for several centuries. The PDO is a main index of major
variations in the North Pacific climate and ocean productivity (Mantua et al., 1997; Jacoby et al., 2004). In particular,
according to instrumental data analysis (Shatilina, Anzhina, 2008), the last warming of the northern part of the Pacific
Ocean (since 1970s) resulted in the intensive temperature increase and precipitation decrease in the southern part of
the Russian Far East.
We suppose that 9-year cycle may be related to solar activity, as, first of all, many authors showed influence of solar
activity on the climate variability (Bond et al., 2001; Lean et al., 1999; Lean, 2000; Mann et al., 2009; Zhu et al.,
2016). Secondly, the significant correlation between of the August-December minimum temperature reconstruction
and TSI can be regarded as an additional evidence of this assumption. And, finally, there is a coincidence of the
reconstructed cold periods with the Maunder Minimum (1645–1715) and the Dalton minimum period centered in
1810. The solar activity influence in the region is traditionally associated with an indirect effect on the circulation of
the atmosphere (Erlykin et al., 2009; Fedorov et al., 2015). In the second half of the 20th century the solar radiation
intensity changes contributed to more intensive warming of the equatorial part of the Pacific Ocean and more active
inflow of warm air masses to the north (Fedorov et al., 2015).
In spite, the fact that it is quite difficult to reveal for certain long-period cycles in a 486-year chronology, we
nonetheless revealed the 189-year cycle (MTM) or 200-year cycle (SSA analysis), which probably may possibly be
linked to the solar activity. Close periodicity is revealed in long-term climate reconstructions and is linked to the
quasi-200-year solar activity cycle in other study (Raspopov et al., 2008; Raspopov et al., 2009). Raspopov et al.
(2008) showed that in tree-ring based reconstructions the cycle varies from 180 to 230 years. Moreover, the high
correlation between the minimum temperatures reconstruction and TSI and also the revealed link between the
reconstructed temperatures and solar activity minima lead to suppose that the solar activity may be the driver of the
200-year cycle.  Such climate cycling, linked not only to temperature but also to precipitation, is revealed for the
territories of Asia, North America, Australia, Arctic and Antarctic (Raspopov et al., 2008). At the same time, the 200-
year cycle (*de-Vries* cycle) may often have a phase shift from some years to decades and correlates not only positively
but also negatively with climatic fluctuations depending on the character of the nonlinear response of the atmosphere-
ocean system within the scope of the region (Raspopov et al., 2009). According to Raspopov et al. (2009), the study
area is in the zone that reacts with a positive correlation to solar activity, though the authors note that we should not
expect a direct response because of the nonlinear character of the atmosphere-ocean system reaction to variability in
solar activity (Raspopov et al., 2009). Taking into consideration this fact and that the cold and warm periods shown
in our reconstruction are slightly shifted compared with more continental areas and the whole Northern Hemisphere,
we can say that the reconstruction of minimum August-December temperatures reflects the global climate change
process in aggregate with the regional characteristics of the study area.
**Conclusions**
Using the tree-ring width of *Pinus koraiensis,* the mean minimum temperature of the previous August-December has
been reconstructed for the southern part of Sikhote-Alin Mountain Range, northeastern Asia, Russia, for the past 486
years. This dataset is the first climate reconstruction for this region, and for the first time for northeast Asia, we present
a reconstruction with a length exceeding 486 years.
Because explained variance of our reconstruction is about 39%, we believe that the result is noteworthy as it displays
the respective temperature fluctuations for the whole region, including northeast China, the Korean peninsula and the
Japanese archipelago. Our reconstruction is also in good agreement with the climatic reconstruction for the whole
Northern Hemisphere. The reconstruction shows good agreement with the cold periods described by documentary
notes in eastern China and Japan. All these comparisons prove that for this region, the climatic reconstruction based
on tree-ring chronology has a good potential to provide a proxy record for long-term, large-scale past temperature
patterns for northeast Asia. The results show the cold and warm periods in the region, which are conditional on global
climatic processes (PDO), and may reflect the influence of solar activity (the 9-11-year and 200-year solar activity
cycles). At the same time, the reconstruction highlights the peculiarities of the flows of global process in the study
area and helps in understanding the processes in the southern territory of the Russian Far East for more than the past
450 years. Undoubtedly, the results of our research are important for studying the climatic processes that have occurred
in the study region and in all of northeastern Asia and for situating them within the scope of global climatic change.

**Acknowledgements** This work was funded by Russian Foundation for Basic Research, Project 15-04-02185.

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

**Tables**
**Table 1.** The sampling information and statistics of the signal-free chronology

| | VUSr |
|---|---|
| Elevation (m a.s.l.) | 700-900 |
| Latitude (N), Lingitude (E) | 44°01'32'', E 134°13'15'' |
| Core (live trees) / sample (dead trees) | 25/20 |
| Time period / length (year) | 1451-2014 / 563 |
| MS | 0.253 |
| SD | 0.387 |
| AC1 | 0.601 |
| R | 0.691 |
| EPS | 0.952 |
| Period with EPS>0.85 / length (year) | 1602-2014 / 412 |
| Period with EPS>0.75 / length (year) | 1529-2014 / 485 |
| Skew/Kurtosis | 0.982/5.204 |

MS – mean sensitivity, SD – standard deviation, AC1 – first-order autocorrelation, EPS – expressed population signal

**Table 2.** Calibration and verification statistics of the reconstruction equation for the common period 1971-2003 of
Bootstrap

| Statistical item | Calibration | Verification (Bootstrap, 199 iterations) |
|---|---|---|
| r | 0.62 | 0.62 (0.54-0.70) |
| $R^2$ | 0.39 | 0.39 (0.37-0.41) |
| $R^2_{adj}$ | 0.36 | 0.37 (0.37-0.40) |
| Standard error of estimate | 1.20 | 1.11 |
| F | 18.76 | 18.54 |
| P | 0.0001 | 0.0001 |
| Durbin-Watson | 1.73 | 1.80 |


**Table 3.** Cold and warm periods based on the results of this study compared with other researches

| Period | Southern Sikhote-Alin (this study) | Laobai Mountain (Lyu et al., 2016) | Changbai Mountain (Zhu et al., 2009) |
|---|---|---|---|
| Cold | 1535-1540[1]; 1550-1555[1] | * | * |
| | — | 1605-1616 | |
| | 1643-1649; 1659-1667 | 1645-1677 | * |
| | 1675-1689 | 1684-1691 | * |
| | 1791-1801; 1807-1818 | — | 1784-1815 |
| | 1822-1827; 1836-1852 | | 1827-1851 |
| | 1868-1887 | — | 1878-1889 |
| | 1911-1925 | 1911-1924; 1930-1942; 1951-1969 | 1911-1945 |
| Warm | 1560-1585[1] | * | * |

| | | |
|---|---|---|
| 1600-1610[1]; 1614-1618 | — | * |
| 1738-1743 | — | — |
| 1756-1759; 1776-1781 | 1767-1785 | 1750-1783 |
| *1787-1793*** | 1787-1793 | — |
| *1795-1807*** | 1795-1807 | — |
| *1855-1865*** | — | 1855-1877 |
| 1944-2014 | 1991-2008 | 1969-2009 |

Note: *italic* ** – the periods which agreement with VUSr but not reliably for VUSr; * - the reconstruction not
covering this period; [1] – uncertain periods, when chronology has EPS> 0.75 (AD 1529-1609).

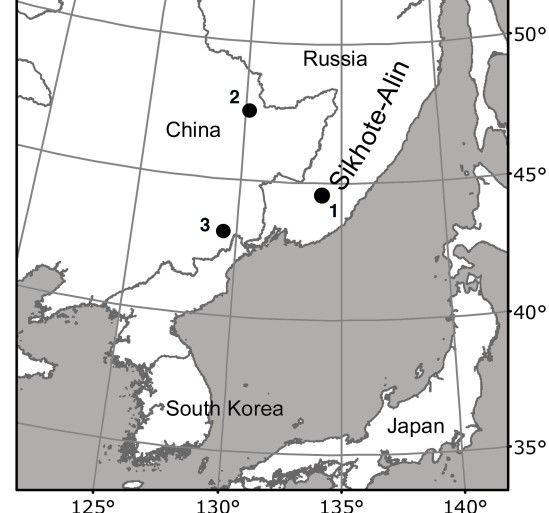


**Figure 1:** Location of the study area on the Sikhote-Alin Mountains, Southeastern Russia (1) and sites of compared
temperature reconstructions: April – July minimum temperature on Laobai Mountain by Lyu et al., 2016 (2), and
February – April temperature established by Zhu et al. (2009) on Changbai Mountain (3).

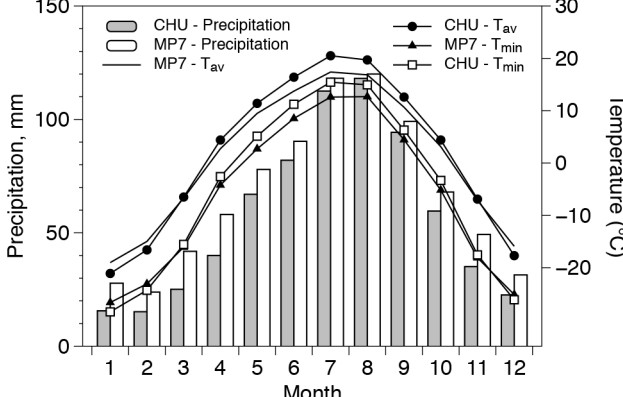

**Figure 2:** Mean monthly (1936-2004), minimum temperature (1971-2003) and total precipitation (1936-2004) at
Chuguevka and mean monthly, minimum temperature and total precipitation for VUS meteorological station (MP7)

627    (1966-2000)

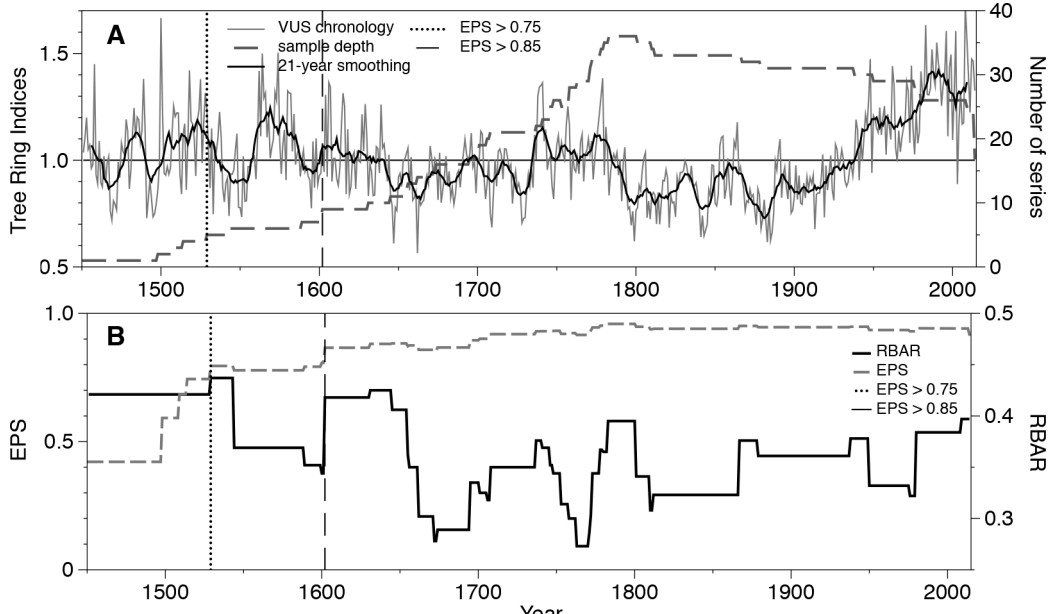


**Figure 3:** Variations of the VUS chronology and sample depth (a) and the expressed population signal (EPS) and
average correlation between all series (Rbar) VUS chronology from AD 1451 to 2014 (b)

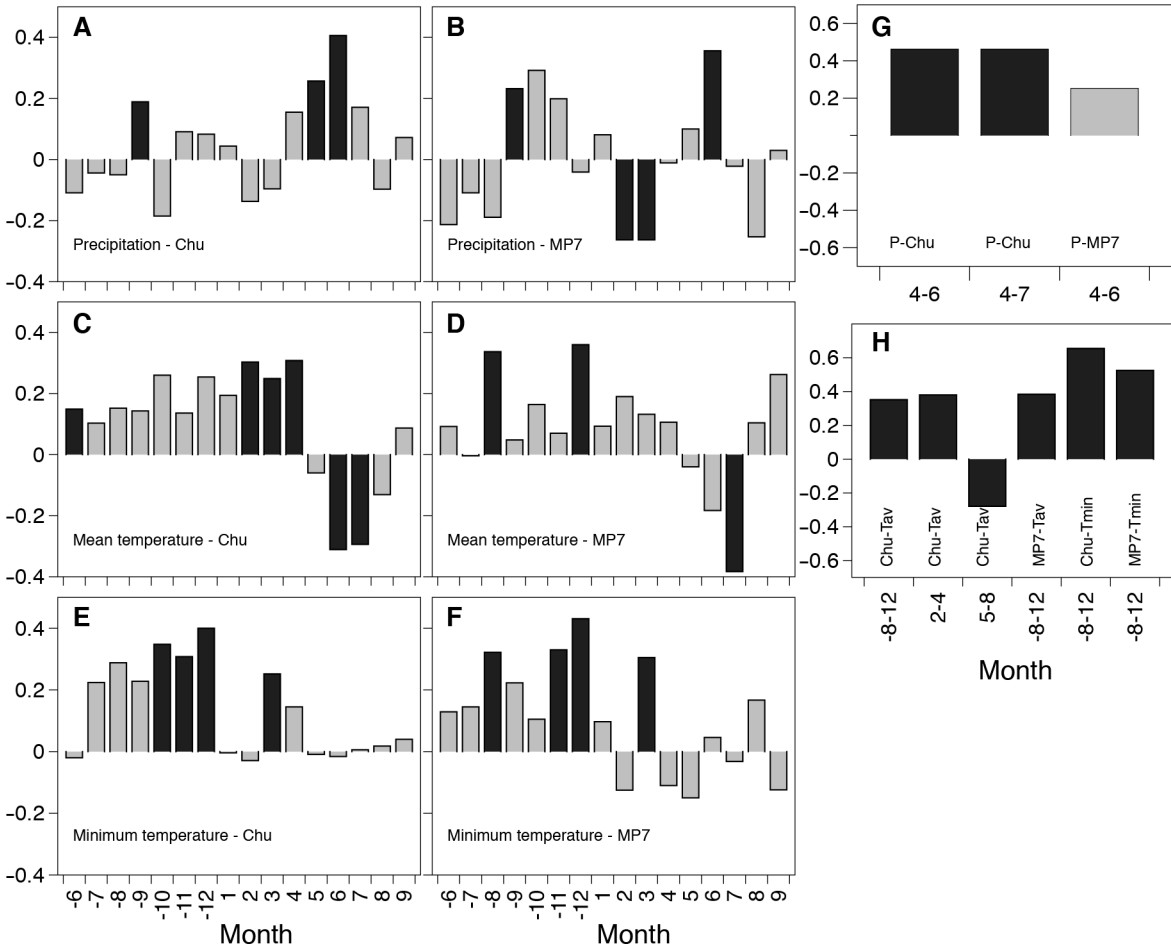

Month
**Figure 4:** Correlations between the monthly mean meteorological data and VUS chronology
A, C, E – Chuguevka (Chu) and VUS chronology; B, D, F - VUS meteorological station (MP7) and VUS chronology;
G – correlation coefficients between VUS chronology and the precipitation of different month combinations; H –
correlation coefficients between VUS chronology and the temperature of different month combinations. The black
bars are significant value.

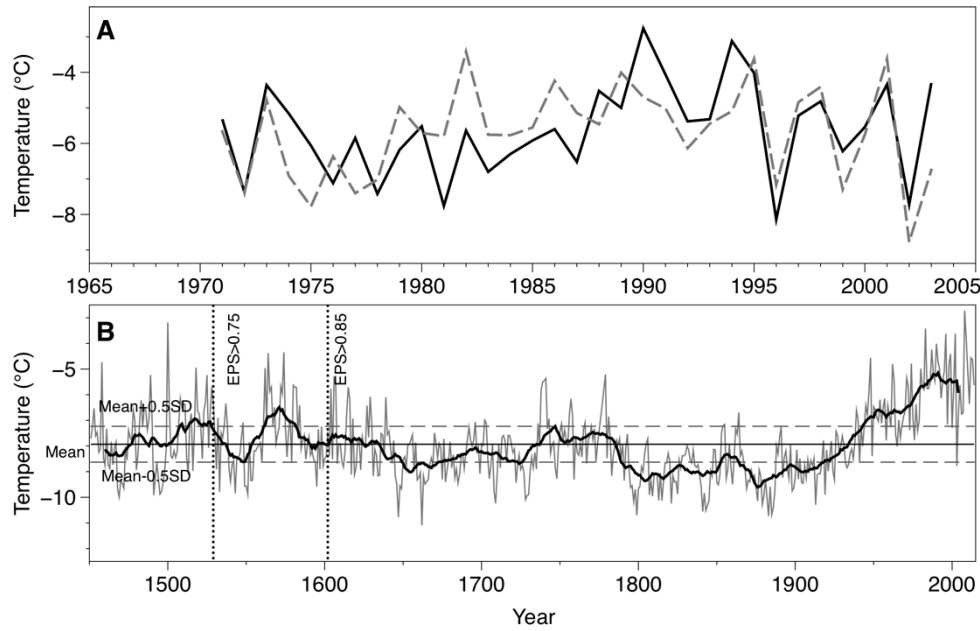


**Figure 5:** (a) Actual (black line) and reconstructed (dash line) August – December minimum temperature for the common period of 1971-2003; (b) reconstruction of August – December minimum temperature (VUSr) to Southern part of Sikhote-Alin for the last 563 years. The smoothed line indicates the 21-year moving average.

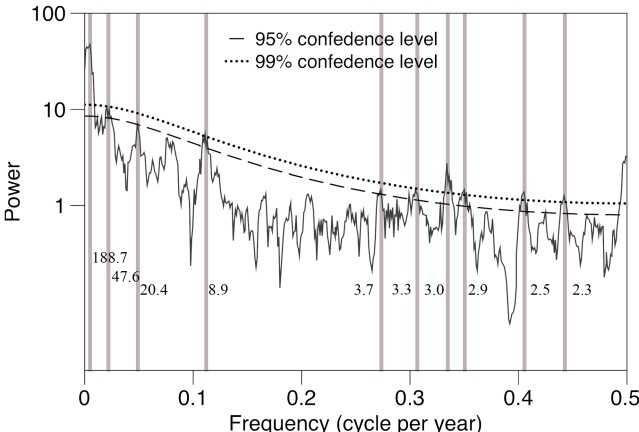

**Figure 6:** The MTM power spectrum of the reconstructed August – December minimum temperature (VUSr) from 1529 to 2014

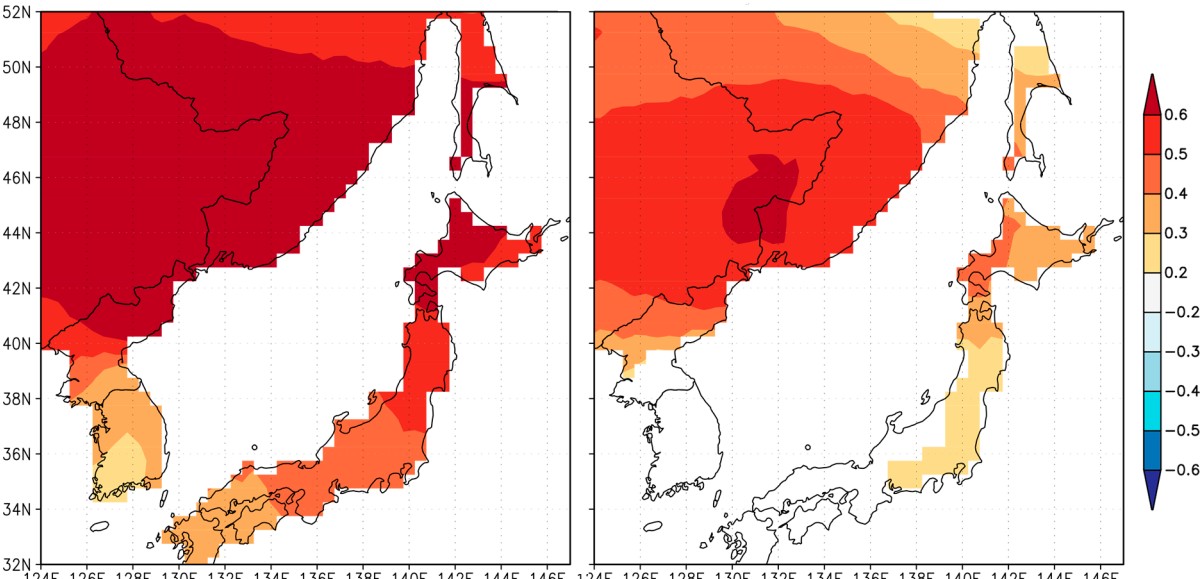

**Figure 7:** Spatial correlations between the observed (a) and reconstructed (b) August – December minimum temperature (VUS) in this study and regional gridded annual minimum temperature from CRU TS 4.00 over their common period 1960–2003 ($p < 10\%$).


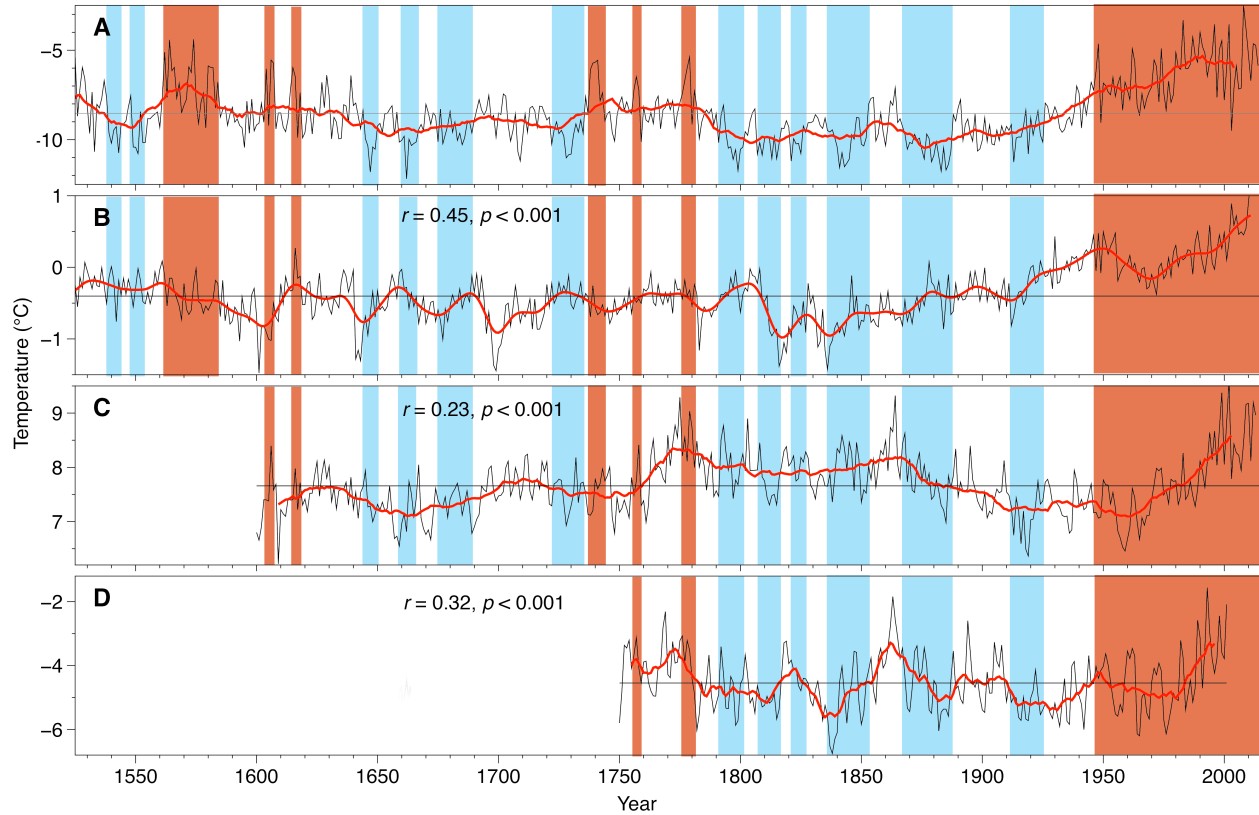


**Figure 8:** (a) August-December mean minimum temperature reconstructed (VUSr) on southern part of Sikhote-
Alin, (b) Northern Hemisphere extratropical temperature (Willes et al., 2016), (c) April – July minimum temperature
on Laobai Mountain by Lyu et al., 2016, and (d) February – April temperature established by Zhu et al. (2009) on
Changbai Mountain. Black lines denote temperature reconstruction values, and red color lines indicate the 21-year
moving average; red and blue fields – warm and cold period consequently (in this study)