# Peer review of "Autumn – winter minimum temperature changes in the"

_Climate of the Past, 2017_

## Referee Comment (RC1) · Anonymous Referee #1 · 20 Sep 2017

General Comments: In this study, a Pinus koraiensis tree-ring chronology at the southern Sikhote-Alin mountain range of northeast Asia was used to reconstruct an autumnwinter minimum temperature spanning the period 1509-2015. Temperature reconstructions are rare in this region and this reconstruction is very valuable for the supplement of the local temperature series. However, some issues existed in this reconstruction. This biggest problem of this study is that the explained variance of the reconstruction equation is very low. The low explanation means the reliability of the reconstruction equation decreases. In addition, the year to year (high-frequency) variations of the reconstructed series was not in good agreement with the actual minimum temperature series (Fig. 5a). The correlation (0.52) may be caused by the similar trends. Thus, the

real correlation coefficient between tree-ring index and autumn-winter minimum temperature might be lower than 0.52, which could be tested by calculating the 1st-order difference correlation coefficient between them. Please try using some methods to increase the amount of the explanation of the reconstruction equation. In addition, the greatest advantage of this reconstruction is that it spans a longer time range (more than 500 years), which can capture low-frequency climate variations (as the author said in Lines 40-48, 51). We know it is very important to extend the reconstruction series (or tree-ring chronology), but a generally acceptable threshold of the EPS is greater than 0.85. However, the EPS value from AD 1509 to 1602 is only greater than 0.7 and it contained 3 trees (or cores) (lines 119-124). Please try to find more older trees if you want to make up for this deficiency. Therefore, I cannot recommend it is accepted to publish in the current version.

Specific Comments: 1. Five main objectives of this study are two much. The objectives (1) and (2) that develop the first (more than 500-year) tree-ring-width chronology in the far eastern region are not the real objectives. Please only list the most important goals and make them less than three. 2. It's impressive that the authors say "two cores per undamaged old-growth mature tree (50 cores from 25 trees) and one sample from dead trees (20 samples) were extracted from Pinus koraiensis trees in the sample plots ......" (lines 98-99). However, the maximum sample depth of the VUS chronology shown in Figure 3 is nearly 35. It is far less than the actual sample depth. Please check this inconformity or give a reasonable explanation. 3. The reconstruction period of this study is from 1509 to 1602, which matches the EPS > 0.7, while the authors highlighted the EPS with the value 0.75 in figure 3. Please let them keep consistent. 4. Some figures (for example, Fig. 3, 5, 8) in the manuscript have no Y-axis title. Please add it. 5. In the manuscript, new plant name should be added with Latin name only if it appears for the first time. Please write the whole Latin name, for example the P. koraeinsis in line 20, and the A. nephrolepis, B. costata, P. jezoensis, P. koraiensis, and T. amurensis (lines 79-80). 6. Two climate data sets (Chuguevka and MP7) were used to evaluate the tree growth-climate relationships, but in Figure 2 only

the climate (monthly temperature and total precipitation) of MP7 meteorological station were shown. It is better to add the data of another weather station. 7. Why there are some big difference in the results of tree growth-climate relationships between long (Chuguevka) and short (MP7) climate data sets? Is it because the tree growth-climate relationships are unstable over time? If it is, the tree-ring data might be not suitable for the climate reconstruction. 8. There are some methodological and results sentences in discussion section, please move them into the correct places (method or result section), such as lines 268-269, lines 349-351. 9. There are some Russian in line 275, please change them to English.

Please also note the supplement to this comment:
https://www.clim-past-discuss.net/cp-2017-98/cp-2017-98-RC1-supplement.pdf
* * *

---

## Author Comment (AC1) · 7 Oct 2017

Dear Referee, We carefully revised our manuscript according to your comments. We very appreciate your helpful comments on our manuscript. These comments help as to make our reconstruction more perfect and accurate. All detailed revision and response are as below. Thank you so much for all your help. To our response we attached a file in which all the changes in the main text of the manuscript are highlighted in yellow.

Sincerely yours, Olga Ukhvatkina and co-authors

Response to general comments:

Comment: "This biggest problem of this study is that the explained variance of the reconstruction equation is very low. The low explanation means the reliability of the reconstruction equation decreases. In addition, the year to year (high-frequency) variations of the reconstructed series was not in good agreement with the actual minimum temperature series (Fig. 5a). The correlation (0.52) may be caused by the similar trends. Thus, the real correlation coefficient between tree-ring index and autumn-winter minimum temperature might be lower than 0.52, which could be tested by calculating the 1st-order difference correlation coefficient between them. Please try using some methods to increase the amount of the explanation of the reconstruction equation."

The authors' response: We are very grateful to you for this comment. We once again tried to search for available meteorological data and found data on the minimum temperatures from Chuguevka. Then we tried to reconstruct the minimum temperatures (August-December for the previous year) based on these data. This allowed us to significantly increase the explained variance to 39% (our previous result was only 25%). Similar values are often found for reconstruction in the East-Asian region (e.g. Willes et al., 2014). Interestingly, the temperature correlation between Chuguyeva and MP7 for all months is 99%, but the correlation for August-December is only 83%. Compared to the data from Chuguyevka, the data from the MP7 for these months looks a little bit "noisy". We think that this is because on MP7 obtaining data in the winter months was sometimes difficult because this is a weather station on a Research Station without permanent staff of meteorologists.

Comment: "In addition, the greatest advantage of this reconstruction is that it spans a longer time range (more than 500 years), which can capture low-frequency climate variations (as the author said in Lines 40-48, 51). We know it is very important to extend the reconstruction series (or tree- ring chronology), but a generally acceptable threshold of the EPS is greater than 0.85. However, the EPS value from AD 1509 to 1602 is only greater than 0.7 and it contained 3 trees (or cores) (lines 119-124). Please try to find more older trees if you want to make up for this deficiency."

The authors' response: Thank you very much for the suggestion. Indeed, the EPS value becomes greater than 0.85 after 1602. But we think that it is very important to extend the reconstruction as far as possible, since there are few long climate reconstructions in this region. Moreover, the northern Hemisphere temperature series (D'Arrigo et al., 2006) and historical documents very rare for the North-East Asia (and are absent for the Russian Far East), confirmed that the reconstruction temperature from 1529 to 1602 is valuable. Therefore, we kept part of the reconstruction from 1529 to 1602 (EPS>0.75). In order for the reader to understand this, we added some clarifications to the text about the EPS value from 1529 to 1602, and also added lines denoting this part on Fugures 3 and 5.

Response to specific comments:

"1. Five main objectives of this study are two much. The objectives (1) and (2) that develop the first (more than 500-year) tree-ring-width chronology in the far eastern region are not the real objectives. Please only list the most important goals and make them less than three."

The authors' response: Comment accepted. We reduced number of objectives from five to three.

"2. It's impressive that the authors say "two cores per undamaged old-growth mature tree (50 cores from 25 trees) and one sample from dead trees (20 samples) were extracted from Pinus koraiensis trees in the sample plots ......" (lines 98-99). However, the maximum sample depth of the VUS chronology shown in Figure 3 is nearly 35. It is far less than the actual sample depth. Please check this inconformity or give a reasonable explanation."

The authors' response: It was not a mistake. In fact, we took 2 cores from each living tree, but only one was used for analysis - with a large number of tree rings (we did not mention this in the text). Thus, the total number of samples in the analysis was 45. In addition, we do not have time periods when all 45 samples overlap with each other because some old trees died before new live ones appeared. Therefore, the maximum number of samples on the Figure 3 is less than 40. We made changes to the text, so as not to confuse the reader.

"3. The reconstruction period of this study is from 1509 to 1602, which matches the EPS > 0.7, while the authors highlighted the EPS with the value 0.75 in figure 3. Please let them keep consistent."

The authors' response: Comment accepted. We deleted line denoting EPS>0.7 (fig. 3 and fig. 5) and added lines denoting EPS>0.75 and EPS>0.85 on Figures 3 and 5.

"4. Some figures (for example, Fig. 3, 5, 8) in the manuscript have no Y-axis title. Please add it."

The authors' response: Comment accepted, titles added.

"5. In the manuscript, new plant name should be added with Latin name only if it appears for the first time. Please write the whole Latin name, for example the P. koraeinsis in line 20, and the A. nephrolepis, B. costata, P. jezoensis, P. koraiensis, and T. amurensis (lines 79-80)."

The authors' response: Comment accepted. In the text we added whole Latin names (lines 89 and 90): Abies nephrolepis (Trautv.) Maxim, Betula costata (Trautv.) Regel., Picea jezoensis (Siebold et Zucc.) Carr., Pinus koraeinsis Siebold et Zucc., and Tilia amurensis Rupr.

"6. Two climate data sets (Chuguevka and MP7) were used to evaluate the tree growth-climate relationships, but in Figure 2 only the climate (monthly temperature and total precipitation) of MP7 meteorological station were shown. It is better to add the data of another weather station."

The authors' response: Comment accepted. We added climatic parameters of Chuguevka to fig. 2 too.

"7. Why there are some big difference in the results of tree growth-climate relationships between long (Chuguevka) and short (MP7) climate data sets? Is it because the tree growth-climate relationships are unstable over time? If it is, the tree-ring data might be not suitable for the climate reconstruction."

The authors' response: After we found the data on the minimum temperatures from Chuguevka, we compared this data with MP7. The results showed that the correlation between the data from Chuguevka and VUS for all months is very high (99%). But the correlation between the data in the winter period is significantly reduced (up to 83%) – see also response to general comment 1. Now Figure 4 clearly shows that the results of the interconnections are very similar for these two weather stations and differ only in the degree of severity.

"8. There are some methodological and results sentences in discussion section, please move them into the correct places (method or result section), such as lines 268-269, lines 349-351."

The authors' response: Comment accepted. We added new text to the Methods section (lines 140-143) and also moved and added the comparison results of the reconstruction and CRU TS4.00 in the Results (lines 216-218) and the Discussion (lines 255-261). However, we left the text (lines 349-351) in Conclusion, as we believe that it is important to repeat this here.

"9. There are some Russian in line 275, please change them to English."

The authors' response: Comment accepted.

Please also note the supplement to this comment:
https://www.clim-past-discuss.net/cp-2017-98/cp-2017-98-AC1-supplement.pdf

**Supplement:**

[revised manuscript text omitted]

---

## Referee Comment (RC2) · Anonymous Referee #2 · 11 Oct 2017

General Comments: The authors reconstructed minimum temperature using Pinus koraiensis in the Russian Far East. The paper is structured well. The merit of the manuscript is the longevity of the trees that were sampled, although the fidelity through time drops. I have a few concerns about this paper before I think it can be published. I worry that the reconstruction should cut at 1600, where there is more sample depth and a higher EPS value. Generally, the rule is 0.85 and I've seen others use 0.80 but not 0.75 as the authors do. Further, the sample depth during the period prior to 1600 is very small, less than 5 cores. I also wonder why the authors are comparing their reconstruction with other reconstructions from different seasons. I think there is merit to this paper and think some of my comments could be issues of clarity but would like the

authors to consider them to determine if these are methodological concerns or clarity issues. Specific comments: Line 219: Why do the authors use Aug-Dec when not all the months are significantly correlated? Line 244: The explanation of KNMI needs to be in the methods. Lines 251: This is a bigger point, why are the authors comparing the Aug-Dec min temperature reconstruction to different seasonal reconstructions? This in itself is not wrong but there needs to be some explanation as to why the signals are different in these reconstructions. I'd be more comfortable with different seasonal comparison with the overall NH reconstructions but wonder why the two reconstructions that are 500km and 430km away are from April to July and Feb. to April. This is especially strange when the authors state that others have found this same Aug-Dec signal but do not compare their reconstruction to those. There could be different reasons for seasonal shifts in climate. Thus, I think this needs to be handled carefully. Figure 5: The relationship between tree growth and instrumental temperature looks a little weak. I would like the authors to discuss what the tree-rings are not getting (i.e., peaks or troughs). I also worry that the higher r-value is more of an artifact of both timeseries trending upward rather than a true correlation. Figure 7: I'm not sure why this figure is in here. Are the authors trying to show that region has a strong consistent climate signal? If so, then again why are the other regional reconstructions based off of different seasons? Perhaps I'm missing something due to clarity?

---

## Referee Comment (RC3) · Anonymous Referee #3 · 18 Oct 2017

This study presents a new climate reconstruction based on tree rings from Korean pine trees located in the Russian Far East. The new annually-resolved record, which spans more than five centuries, derives from an area wherefrom such centennial-scale climate reconstructions are sparse. The new record is therefore really exciting and important, as it may help improve our understanding of climate change in this area. Consequently, this study clearly deserves to be published, although after some major revisions.

The authors argue that the main limiting factor for tree-ring growth is the August-December temperature of the year prior to the growth year, implying that the new record documents changes in the August-December temperature in the period 1509-2014 AD. However, establishing the August-December-temperature as the dominant control on tree-ring growth is not straightforward and this aspect is not discussed in sufficiently detail to make it appear robust. Only in the Conclusion section do we learn that the August-December temperature only explains 25% of the variance of the data. Similarly, other important aspects are not discussed in sufficient detail and important information is lacking - these aspects should be addressed in the review process. In general, the paper is reasonably well written, but some aspects/sentences are unclear and the paper would benefit from a thorough check of the language. Below, I list my main concerns and some minor comments to the paper.

Main concerns

**Establishing the dominant control on tree-ring growth**
The paper initially discusses potential climatic parameters as the dominant mechanism controlling tree-ring growth, then – in one sentence – concludes that "the most stable correlation appears between the growth and the minimum monthly temperature of August-December of the previous year at MP7, on which we base our subsequent reconstructions". This aspect is critical, because this is where the meaning of the climate reconstruction is defined, and the treatment of this aspect is too superficial and not sufficiently robust. It is unclear what "stable correlations" refer to, and it should be discussed how much of the variance is actually explained by this parameter – that turns out (in the Conclusion section) to be quite a small fraction. Based on Fig. 4, it seems that several of the other climatic parameters correlate with the tree-ring growth almost as well as the August-December temperature. This should be explored – and discussed - in more detail. Also, would it make sense to use principal component analysis and combine some of the climatic parameters to see if it is possible to explain more of the variance in the data – although this will not make it possible to reconstruct more climate parameters back in time, it may still prove helpful for our understanding of the climate parameters driving tree-ring growth.

It is also unclear how the bootstrapping method was used for the verification – vital details are missing as this is not explained in the text. It just states that: "The idea that this method is based on indicates that the available data already include all the necessary information for describing the empirical probability for all statistics of interest". It is unclear what this actually means, and it should be explained in more detail how the verification is done.

**Defining warm and cold periods**
The occurrence of warm and cold periods in the new record is defined as when the temperature deviates more than half the standard deviation from the mean. However, it is unclear if this refers to the 21-yr smoothed record, or the annual data – my guess is the annual

data, but the text seems to suggests the 21-yr averaged data, and it is impossible to tell from the figure. The problem with the definition and the figure (Fig. 5) is that it is hard to make them match, i.e. the 21-yr smoothed record rarely increase/decrease above/below the dashed lines (or is that because it is the standard deviation on Fig. 5 and not half the standard devation?), whereas the annual data show more variability, briefly extending beyond the standard deviation on many occasions – but in this case the defined periods seem very arbitrary and could as well have been longer or shorter. Also, looking at Fig. 5b, the four warmest years do not occur during the years cited in the text (although it is unclear if this is based on the annual or the 21-yr smoothed data, but neither seem to fit the description in the text).

**Discussion of regional climate variability**
First of all, it is a bit difficult to follow the discussion of regional climate changes without a map, where the location of some of the other records are indicated (e.g. Fig. 7). Secondly, the discussion is somewhat unclear, because it is concluded that "…these results characterize regional climate variations and provide reliable data for large-scale reconstructions for the northeastern portion of Eurasia". At the same time, there are clearly differences between the record presented in this study and the nearby records shown in Fig. 8c and 8d. The differences between the new climate record presented in this study and those from nearby areas are briefly discussed in lines 308-316, and are attributed to differences in the reconstructed temperature parameters – and the asymmetry between medium, minimum, and maximum temperatures. This is a really important aspect, as the different records reconstruct the temperature during different parts of the year, as it is therefore a little like comparing apples and oranges. I think this aspect deserves much more attention, particularly if we are to understand the regional climate variability. It also raises the question as to how and the extent to which we should understand the new record as representing regional climate variability.

**Spectral analysis and links to solar cycles**
First of all, there is no description of the methods underlying the spectral analysis. The paper just states that "The MTM analysis over the full length…", which means that it is impossible to reproduce the spectral results presented in this paper. The Methods section should provide sufficient details of the method used to enable other people to reproduce the results.

Secondly, in the Results and Discussion sections a myriad of significant 2-3 year cycles (2.3, 2.5, 2.9, 3.0, 3.3, and 3.7) are reported and discussed. While these periodicities may be real – and potentially reflect the ENSO or quasi-biennial oscillation – they are very close to the Nyquist frequency. With a Nyquist period of 2 years, it is hard to interpret the 2-3 years as direct evidence for climatic oscillations on this time scale. It is thus likely that these high-frequency periods reflect year-to-year scatter, but this aspect is not discussed as all.

Thirdly, the Abstract and Conclusion mention an 11-year cycle, but the 11-yr cycle is not visible in the power spectrum (Fig. 6), and the Results section only mentions the 8.9-yr cycle, whereas the Discussion section mentions a 8.9-11.5-yr cycle. But where did the 11- or 11.5-yr cycle come from? There is no mention of this and this a confusing.

Finally, a more general criticism of this aspect concerns the discussion of the origin of the periodicities. The main problem is that the periodicities, in particular the 8.9- and 189-yr cycles, uncritically are taken as direct evidence for a strong solar influence on climate on these time scales. While the Sun may have driven climate change on these time scales in the study area, it is simply not enough to infer this based on periodicities that resemble those of the Sun (which on average are 11 and 210 yeas, respectively). In such a record, there will almost always be periodicities that resemble those of the Sun and it therefore takes more to infer causality. In the authors want to establish that the Sun influenced climate in the area, they should engage in much more detailed analysis of the new tree-ring climate record and

records of solar activity and calculate correlations, lads, and compare those to red-noise models. It would also be interesting to establish if the cold period indeed corresponds to solar minima – as stated in the abstract – but such an analysis is completely missing.

**Minor comments**

L. 19. Abstract: It is unclear what you mean by "de-Vier quasi-200 quazi-200 solar activity cycle." Presumably this refers to the de Vries (or Suess) 210-year solar cycle. The word "year" is also missing.

L. 47. "It is well known that warming of the climate is correlated with solar activity". This sentence and the following sentence suggest that solar activity is the only source of warming, including global warming. You need to be much more precise with respect to what you mean here. Also, solar activity is a driver of climate change, but it is not strong driver a temperature changes compared to changes in greenhouse gases.

L. 128-129. Maybe spell out what is meant by "...it matches a minimum sample depth of 3 trees in this segment".

L. 133. Where is the Chuguevka meteorological station relative to the sample site? This is really an import aspect.

L. 175. There is no "Y" in the equation – guess this refers to "VUSr"?

L. 297. "The period of landscape formation……during the transition". This is unclear, as the landscape formation occurred long before the Little Ice Age – do you refer to vegetation changes?

L. 326-327. This sentence makes no sense to me – how is this related to the sentences above (which it refers to)?

**Figures**

**Figure 3**
It is unclear what the sample depth refers to. Is it he number of tree records?

**Figure 7**
It is the correlation coefficient that is plotted here – this is not clear to me? It is also unclear if the signifance refers to all colours, so that for white areas there is no correlation at the 10% signifaince level? It would be very helpful if the geographical position of the record from this study (Fig. 8a) and the two nearby records in Figs. 8c and 8d could be indicated in this plot.

---

## Author Comment (AC2) · 18 Oct 2017

Dear Referee, We sincerely thank you for your careful attitude to our manuscript and your help in improving it. We closely analyzed manuscript in accordance with your comments. These help as to make our research more clarify for readers. All detailed response is below. Thank you for your help!

With best wishes, Olga Ukhvatkina and co-authors.

Response to general comments:

1. Comment: I worry that the reconstruction should cut at 1600, where there is more

sample depth and a higher EPS value. Generally, the rule is 0.85 and I've seen others use 0.80 but not 0.75 as the authors do. Further, the sample depth during the period prior to 1600 is very small, less than 5 cores.

The authors' response: As we have already responded to the Referee #1, in our opinion it is very important to extend the reconstruction as far as possible, since there are few long climatic reconstructions for this region. Therefore, we would like to keep part of the reconstruction from 1529 to 1602. In order for the reader to better understand that from 1529 to 1602 the EPS value is above 0.75 but less than 0.85, we added some clarifications to the text. We also added lines denoting reconstruction part with 0.75< EPS<0.85 in Figures 3 and 5. However, if the Editor decides that this part should be excluded, we will make this change (and, accordingly, changes in further results and conclusions).

2. Comment: I also wonder why the authors are comparing their reconstruction with other reconstructions from different seasons. I think there is merit to this paper and think some of my comments could be issues of clarity but would like the C1 authors to consider them to determine if these are methodological concerns or clarity issues.

Specific comments: Line 219: Why do the authors use Aug-Dec when not all the months are significantly correlated?

The authors' response: Indeed, if we look at individual months, the tree-ring growth is not significantly correlated with the minimum temperature of some of them (in particular, August). However, when we consider combinations of months, the situation changes. We tried all possible combinations of months, before we chose the period from August to December. And if we only consider October-December, the correlation between radial growth and temperature will be 0.56. But if we add August and September, then it rises to 0.62 (new version of manuscript in supplement and the response to Ref.#1).

3. Comment: Line 244: The explanation of KNMI needs to be in the methods.

The authors' response: Comment accepted. (see also response to Ref.#1)

4. Comment: Lines 251: This is a bigger point, why are the authors comparing the Aug-Dec min temperature reconstruction to different seasonal reconstructions? This in itself is not wrong but there needs to be some explanation as to why the signals are different in these reconstructions. I'd be more comfortable with different seasonal comparison with the overall NH reconstructions but wonder why the two reconstructions that are 500km and 430km away are from April to July and Feb. to April. This is especially strange when the authors state that others have found this same Aug-Dec signal but do not compare their reconstruction to those. There could be different reasons for seasonal shifts in climate. Thus, I think this needs to be handled carefully.

The authors' response: As we understood the Referee's comment can be divided into two questions. The first one is why the climatic signal for the same tree species is different though the distances between the study areas are rather small (500 and 400 km). The second one is why we compare the reconstructions for different seasons despite the fact that seasons can have different climatic shifts. First, we will answer the second question. There are few reconstructions for the North-East Asia region, so we have to use the available reconstructions. At the same time, the presence of cold and warm periods generally coincides in the compared reconstructions, while the difference between them can be attributed to different seasonal shifts and local climate specifics (but so far, we say this under correction). We also need to clarify that another two papers which we refer to saying that the other authors have identified similar associations (have revealed the similar correlation) of tree growth and temperature (Line 233) could not be used for comparison. In the first work ("Temperature signals in tree-ring width and divergent growth of Korean pine response to recent climate warming in northeast Asia", Wang et al., 2016), the authors compared the response of Korean pine radial growth to temperature, precipitation and PDSI in different parts of distribution area. The study points in this work were distributed along a latitudinal gradient along the entire boundary area between the northeastern part of China and Russia. The main conclusion of this work was that in different parts of the range there are various limiting factors for the Korean pine growth. However, the authors did not reconstruct the climatic parameters. The second work (Zhang et al., 2015) reveals the response of radial tree growth to the minimum temperatures of August-December of the previous year (as it is in our study), but it was made for the Tibetan plateau. We didn't think it would be correct to compare our results with the results of a study performed on the territory located more than 2.500 km far from the site of our study. The answer to the first question is more complicated. As we have said, the Wang et al., 2016 study shows that the tree response to climatic factors differs in different parts of the range. At the same time, we see that climate in the Sikhote-Alin and Northeast regions of China is very similar, which is also confirmed by Fig. 7a (new version of manuscript). At present, we cannot give the detailed answer to the question of what determines the difference in limiting factors for the Korean pine growth of in different parts of its range. However, we can make some assumptions. To do this, let's compare each neighboring reconstruction with ours separately. The first reconstruction was done by Zang et al. (2016) on the minimum temperatures of April-July. As it's explained by the authors of the article, the warming of the last decades was most strongly expressed by increasing the minimum temperatures in their study area. The authors of this article believe that for their territory the most important limiting factor for the Korean pine growth is the absence of spring and early summer frosts, that allows trees to form wider rings. At the same time, their analysis shows the correlation between the Korean pine growth and the minimal temperatures of August-December of the previous year (as it is in our study). But the correlation for that territory is less significant compared with the minimum temperatures of April-July. We compared the diagram of mean monthly temperature and total precipitation of this article with that one from our study area. The results show that in our study the minimum temperatures are much lower in August-December (especially in October-December), and the minimum temperatures of April-July almost completely coincide in these two points. Probably, that's that affected the differences of limiting factors. As for the second reconstruction (Zhu et al., 2009), the authors of this study did not analyze the minimum temperatures effect on the Korean pine growth. They based their reconstruction at the average monthly temperature of February-April. Fig. 4c of our study clearly shows that it also reveals the correlation between the Korean pine growth and the February-April temperature, but it turned to be lower than the temperature influence in August-December of previous year (Fig. 4h). Perhaps, if the authors of the article Zhu et al., 2009 would have analyzed the correlation of the minimal temperature with the growth of the Korean pine, they could also reveal these relationships. In order for the reader to understand this, we added some clarifications to the text: "Although the spring and summer temperatures have been reconstructed in the last two cases, we use these reconstructions for comparison, because, firstly, there are no other reconstructions for this region, and secondly, despite the possible seasonal shifts, long cold and warm periods should be identified in all seasons". (Lines 265-268).

5. Comment: Figure 5: The relationship between tree growth and instrumental temperature looks a little weak. I would like the authors to discuss what the tree-rings are not getting (i.e., peaks or troughs). I also worry that the higher r-value is more of an artifact of both timeseries trending upward rather than a true correlation.

The authors' response: Comment accepted. According to comment of Referee #1 we improved R-value and R2-value of our reconstruction using more accurate climate dataset (see: response to Ref.#1 and new version of manuscript in supplement).

6. Comment: Figure 7: I'm not sure why this figure is in here. Are the authors trying to show that region has a strong consistent climate signal? If so, then again why are the other regional reconstructions based off of different seasons? Perhaps I'm missing something due to clarity?

The authors' response: Comment accepted. We changed Fig 7. (see new version of manuscript). Fig. 7a showed that minimal temperature in our territories and neighboring territories is very similar. In spite of different response of Korean pine radial growth on climate, Fig. 7b indicating that our temperature reconstruction is representative of large-scale regional temperature variations. But it suggestion needs further researches. We added next text in the manuscript for readers: "We can conclude that the analysis shows that the reconstructed data is representative for large-scale regional temperature variations (Fig. 7). At the same time, some cold and warm periods in our reconstruction and other neighbored studies do not coincide (Fig. 8), which can be due to the reconstruction of other climatic parameters and differing environmental conditions. So, we believe that these results can characterize regional climate variations and provide reliable data for large-scale reconstructions for the northeastern portion of Eurasia, but their use for large-scale regional reconstructions requires further research." (Lines 325-330, new version)

Please also note the supplement to this comment:
https://www.clim-past-discuss.net/cp-2017-98/cp-2017-98-AC2-supplement.pdf

―――――――――――――――――

**Supplement:**

[revised manuscript text omitted]

---

## Author Comment (AC3) · 17 Nov 2017

Dear Referee, Thank you very much for your attention to our research. We closely analyzed manuscript in accordance with your comments and hope that our answers will be satisfactory to you. Some your comments the same with comments of Ref.#1 and Ref.#2 and we ventured repeat our answers to these comments again. We express to you our deep appreciation for your help, which has greatly improved our manuscript. Corrections in the manuscript (new ver.) according to your comments highlighted in green color.

With kind wishes, Olga Ukhvatkina and co-authors.

[Figure]

Response to general comments:

1. Comment: The paper initially discusses potential climatic parameters as the dominant mechanism controlling tree-ring growth, then – in one sentence – concludes that "the most stable correlation appears between the growth and the minimum monthly temperature of August- December of the previous year at MP7, on which we base our subsequent reconstructions". This aspect is critical, because this is where the meaning of the climate reconstruction is defined, and the treatment of this aspect is too superficial and not sufficiently robust. It is unclear what "stable correlations" refer to, and it should be discussed how much of the variance is actually explained by this parameter – that turns out (in the Conclusion section) to be quite a small fraction. Based on Fig. 4, it seems that several of the other climatic parameters correlate with the tree-ring growth almost as well as the August-December temperature. This should be explored – and discussed - in more detail. Also, would it make sense to use principal component analysis and combine some of the climatic parameters to see if it is possible to explain more of the variance in the data – although this will not make it possible to reconstruct more climate parameters back in time, it may still prove helpful for our understanding of the climate parameters driving tree-ring growth.

The authors' response: Comment accepted, "stable correlation" changed to "significant correlation". Analysis of correlation between climatic parameters and tree-ring weight was conduct in specific package for dendroclimatic studies "treeclim" in R (Zang, Biondi, 2014) (reference in main text: Lines 155-156). This is citation of package authors (Zang, Biondi, 2014): Numerically, treeclim uses the algorithm implemented in DENDROCLIM2002 to calculate response and correlation functions; format of input data is the same as for DENDROCLIM2002 and bootRes. In the case of response functions, the design matrix is orthogonalized so that the regression is performed against principal components of the design matrix, retained according to the PVP criterion (Guiot 1991), which corresponds to the determinant of a correlation matrix of uncorrelated variables. Estimated regression coefficients are then transformed back into the

original parameter space (Zang and Biondi 2013). Correlation function analysis uses Pearson's linear correlation computed between the response variable and each sub-vector of the climate design matrix. Bootstrap resampling (1000 iterations) is used to test for significant correlations. This citation is showed that significance of revealed correlations is corroborating by bootstrap resampling analysis (1000 iterations). These methods of analysis are common and "classical" in tree-ring studies. The relatively low value of the explained variance was also noted by Ref.#1 and Ref.#2. According to these comments we improved our reconstruction (see new ver. of manuscript and response to Ref.#1). In additional the R2 value now indicated not only in Conclusion, but also in the section in 3.4 (line 188) and in Table 2. Indeed, the principal component analysis could increase the explained variance, but it is practically not used in such studies, because, as you mentioned, it will not possible to reconstruct climatic parameters.

2. Comment: It is also unclear how the bootstrapping method was used for the verification – vital details are missing as this is not explained in the text. It just states that: "The idea that this method is based on indicates that the available data already include all the necessary information for describing the empirical probability for all statistics of interest". It is unclear what this actually means, and it should be explained in more detail how the verification is done.

The authors' response: Comment accepted. Bootstrap method is the one of the most well-known methods of short data analysis in the tree-ring based studies. Since this method is widely used it is well described in the literature. In main text of manuscript (line 196) we added references, so readers can study the features of the method. In Table. 2 it is indicated that 199 iterations have been carried out for the verification, and in the methodology (Section 2.4) there is a reference to the STATISTICA software we used for the analysis.

2. Comment: Defining warm and cold periods The occurrence of warm and cold periods in the new record is defined as when the temperature deviates more than half the standard deviation from the mean. However, it is unclear if this refers to the 21-yr smoothed record, or the annual data – my guess is the annual data, but the text seems to suggests the 21-yr averaged data, and it is impossible to tell from the figure. The problem with the definition and the figure (Fig. 5) is that it is hard to make them match, i.e. the 21-yr smoothed record rarely increase/decrease above/below the dashed lines (or is that because it is the standard deviation on Fig. 5 and not half the standard devation?), whereas the annual data show more variability, briefly extending beyond the standard deviation on many occasions – but in this case the defined periods seem very arbitrary and could as well have been longer or shorter. Also, looking at Fig. 5b, the four warmest years do not occur during the years cited in the text (although it is unclear if this is based on the annual or the 21-yr smoothed data, but neither seem to fit the description in the text).

The authors' response: Comment accepted. It's our omission that we didn't describe the process of defining of cold and warm periods. In order to clarify this in the main text the following explanation was inserted: "If the reconstructed minimum temperatures were above or below the average value by >0.5 SD for three or more years, then we considered this deviation as warm or cold period, respectively. Also, if two warm (or cold) periods were separated by one year, when the temperature sharply decreased (or increased), then such periods merged into one." (lines 211-214).

3. Comment: First of all, it is a bit difficult to follow the discussion of regional climate changes without a map, where the location of some of the other records are indicated (e.g. Fig. 7). Secondly, the discussion is somewhat unclear, because it is concluded that "...these results characterize regional climate variations and provide reliable data for large-scale reconstructions for the northeastern portion of Eurasia". At the same time, there are clearly differences between the record presented in this study and the nearby records shown in Fig. 8c and 8d. The differences between the new climate record presented in this study and those from nearby areas are briefly discussed in lines 308-316, and are attributed to differences in the reconstructed temperature parameters – and the asymmetry between medium, minimum, and maximum temperatures. This is a really important aspect, as the different records reconstruct the temperature during different parts of the year, as it is therefore a little like comparing apples and oranges. I think this aspect deserves much more attention, particularly if we are to understand the regional climate variability. It also raises the question as to how and the extent to which we should understand the new record as representing regional climate variability.

The authors' response: As we understood this comment may be divided on two parts. First of all, it is necessary to understand where the study areas for which reconstructions being compared. For this we added locations points on the Fig. 7. The next part of the comment concerns the irrelevance of comparing of reconstructions for different seasons. As we answered to Ref.#2 in part this is a fair comment, but we had to use such different reconstructions (see response to com. 4 to Ref.#2) and it is make sense. In addition, we improved the Fig. 7 and now it shows that our reconstruction is representative to the territory of all three reconstructions (for minimum temperature of August - December). Also, despite the fact that the temperature was reconstructed for different seasons, the general trend (cold and warm periods) coincide.

4. Comment: First of all, there is no description of the methods underlying the spectral analysis. The paper just states that "The MTM analysis over the full length...", which means that it is impossible to reproduce the spectral results presented in this paper. The Methods section should provide sufficient details of the method used to enable other people to reproduce the results.

The authors' response: Comment accepted. We added links to the authors of the method and information about used software (lines 156-159).

5. Comment: Secondly, in the Results and Discussion sections a myriad of significant 2-3 year cycles (2.3, 2.5, 2.9, 3.0, 3.3, and 3.7) are reported and discussed. While these periodicities may be real – and potentially reflect the ENSO or quasi-biennial oscillation – they are very close to the Nyquist frequency. With a Nyquist period of 2 years, it is hard to interpret the 2-3 years as direct evidence for climatic oscillations on this time scale. It is thus likely that these high- frequency periods reflect year-to-year scatter, but this aspect is not discussed as all.

The authors' response: We used the additional analysis method (SSA) to confirm the significance of the detected cycles. As a result, we obtained that all 2-3-year cycles are joined in one 3-year cycle. Traditionally, such short-period fluctuations in the region are associated with ENSO or quasi-biennial oscillation and we indicate this in the text. But additional analysis using the KNMI Climate Explorer (http://climexp.knmi.nl) did not reveal a significant correlation between the ENSO indexes and the reconstructed temperatures, but showed a significant correlation with the North Pacific temperature. Therefore, we assume that the Pacific Decadal Oscillation is more important for climate variations. According to the comment and the new results obtained, we made corrections to the main text of the article (lines 139-142, 156-159, 224-233, 363-367, 370-375).

6. Comment: Thirdly, the Abstract and Conclusion mention an 11-year cycle, but the 11-yr cycle is not visible in the power spectrum (Fig. 6), and the Results section only mentions the 8.9-yr cycle, whereas the Discussion section mentions a 8.9-11.5-yr cycle. But where did the 11- or 11.5-yr cycle come from? There is no mention of this and this a confusing.

The authors' response: Comment accepted. We made changes to the manuscript in accordance with this comment, comment #5 and new results obtained. According to an earlier study (Zhu et al., 2016), the 11-year cycle of solar activity in tree-ring reconstructions can be detected as a 8.5-11.5-year.

7. Comment: Finally, a more general criticism of this aspect concerns the discussion of the origin of the periodicities. The main problem is that the periodicities, in particular the 8.9- and 189-yr cycles, uncritically are taken as direct evidence for a strong solar influence on climate on these time scales. While the Sun may have driven climate change on these time scales in the study area, it is simply not enough to infer this based on periodicities that resemble those of the Sun (which on average are 11 and 210 yeas, respectively). In such a record, there will almost always be periodicities that resemble those of the Sun and it therefore takes more to infer causality. In the authors want to establish that the Sun influenced climate in the area, they should engage in much more detailed analysis of the new tree-ring climate record and records of solar activity and calculate correlations, lads, and compare those to red-noise models. It would also be interesting to establish if the cold period indeed corresponds to solar minima – as stated in the abstract – but such an analysis is completely missing.

The authors' response: Comment accepted. We agree that the identification of similar cycles cannot be a direct evidence of the influence of solar activity on the tree growth. Also, the correlation of solar activity indicators with reconstructed temperatures is also not a direct evidence of this. For a full answer to this question, more in-depth studies are needed that, to our opinion, go beyond the scope of this article. However, studies by other authors (e.g., Raspopov et al., 2008) indicate that both short-period and long-period solar activity cycles are directly tracked in tree-ring records and we base our research on these studies. As for comparison of the reconstructed temperatures with the solar activity minimums, we performed an analysis of relationship between our reconstruction and TSI using the KNMI Climate Explorer (http://climexp.knmi.nl). As a result, we obtained a significant correlation with this indicator. In addition, there is analysis of individual cold periods at the end of the 17th century and historical records for neighboring regions in the main text of the manuscript (lines 294-298).

Minor comments L. 19. Abstract: It is unclear what you mean by "de-Vier quasi-200 quazi-200 solar activity cycle." Presumably this refers to the de Vries (or Suess) 210-year solar cycle. The word "year" is also missing. The authors' response: Comment accepted.

L. 47. "It is well known that warming of the climate is correlated with solar activity".

This sentence and the following sentence suggest that solar activity is the only source of warming, including global warming. You need to be much more precise with respect to what you mean here. Also, solar activity is a driver of climate change, but it is not strong driver a temperature changes compared to changes in greenhouse gases. The authors' response: Comment accepted. Line 49.

L. 128-129. Maybe spell out what is meant by "...it matches a minimum sample depth of 3 trees in this segment". The authors' response: This is common expression that mean a number of samples in this part of tree-ring chronology. Usually this expression doesn't need explanation.

L. 133. Where is the Chuguevka meteorological station relative to the sample site? This is really an import aspect. The authors' response: Comment accepted. Line 135.

L. 175. There is no "Y" in the equation – guess this refers to "VUSr"? The authors' response: Comment accepted. Line 188.

L. 297. "The period of landscape formation......during the transition". This is unclear, as the landscape formation occurred long before the Little Ice Age – do you refer to vegetation changes? The authors' response: Comment accepted. Line 330.

L. 326-327. This sentence makes no sense to me – how is this related to the sentences above (which it refers to)? The authors' response: Comment accepted. We rewrote the sentence (L380-382).

Figures Figure 3 It is unclear what the sample depth refers to. Is it he number of tree records? The authors' response: Indeed, this is the common designation of the number of samples.

Figure 7 It is the correlation coefficient that is plotted here – this is not clear to me? It is also unclear if the signifance refers to all colours, so that for white areas there is no correlation at the 10% signifaince level? It would be very helpful if the geographical position of the record from this study (Fig. 8a) and the two nearby records in Figs. 8c and 8d could be indicated in this plot. The authors' response: Comment accepted. We added locations on the Fig 7. As indicated in the caption to this figure, it shows the significant value of the correlation coefficient between our data (instrumental observations - Fig. 7a, reconstruction - Fig. 7b) and model calculated temperatures of the earth's surface (CRU TS 4.00).

Please also note the supplement to this comment:
https://www.clim-past-discuss.net/cp-2017-98/cp-2017-98-AC3-supplement.pdf

―――――――――――――――――――――

[Figure]

**Supplement:**

[revised manuscript text omitted]

---

## Editor Decision (ED1)

Editor's decision for: **Autumn – winter minimum temperature changes in the southern Sikhote-Alin mountain range of northeast Asia since 1509 AD" by Olga N. Ukhvatkina et al.**

Dear Dr. Ukhvatkina,

Thanks you for submitting your manuscript for consideration to "Climate of the Past". As you are aware we have now received three referee comments and after carefully reading through your manuscript, all review comments and your answers, and I am pleased to accept your manuscript with minor revisions.

You have answer all reviewers' comments; but I have below some remarks to your revision and comments.

Both reviewer 1 and 2 suggests that you delete the oldest part of the record where only few data series are available. I agree with you that extending the record as far back in time as possible is valuable. However, here it is imperative to make the readers aware about the significant uncertainty for this older part of the record. You may solve this by adding "Although the record from AD 1529 to 1602 is thus less certain, we here report it as it is very important to extend …" to Line 130 (AR3 version of the ms). Adding this sentence will make it clear to the reader that they need to be more careful when referring to data from the AD 1529-1602 interval.

In relation to this, please also provide information on the uncertainly of the chronology as number of years. E.g., if you mention a cool event from AD 1538-1543, could it for instance be AD 1535-1540 instead? Explaining this will make it easier for non-tree ring specialists to understand the data certainty.

Line 369: You state that the 20-year cycle reflects the PDO. You cannot be sure about this despite the arguments that you present in the following sentence, so please add an "likely", "we suggest" or similar to this sentence. Please also explain how the PDO would influence climate at your study site (temperature, precipitation) though comparison to modern conditions. 1-2 sentences should be sufficient.

I agree with Reviewer 3 that your correlation to solar irradiation cycles is not strong. It is OK to mention the possibility, but firstly you need to 1) make your statements less categorical, making it clear that you suggest the link between solar irradiation and your record due to the comparable timings; 2) provide a short explanation to the mechanism on how changes in solar irradiation would influence climate at your site. The arguments that your provide line 377 and forward that previous papers have shown a link between solar irradiation and climate (mainly at different time scales) is not sufficient evidence for a similar link in your record.

For the 9 year cycle, it is somewhat different than the 11 year solar cycle, and if your chronology is precise is may be a problem. However, the solar cycle is not fully stable, and it is possible that the link is real. Did you make a direct comparison between the instrumental data of the solar irradiation and your data? If not, there is no way that you can be sure that the 9-year cyclicity is linked to solar irradiation.

Also the 189 years cycle in your data is quite far from the 210-year solar cysle, if your chronology is precise. Furthermore, as you acknowledge yourselves, calculating a multi-centennial cyclicity of a record of 486 years is not convincing. So please moderate and tone down your suggested correlation, both in the discussion and the conclusion/abstract.

Language: please check the language of the section lines 369-400, where you have added new text.

Please make sure that all the sites that you mention in the text are provided on your location figure 1.